

# Human milk microbiota associated with early colonization of the neonatal gut in Mexican newborns

Karina Corona-Cervantes[1,*], Igrid García-González[1,*],
Loan Edel Villalobos-Flores[1], Fernando Hernández-Quiroz[1],
Alberto Piña-Escobedo[1], Carlos Hoyo-Vadillo[2], Martín Noé Rangel-Calvillo[3] and
Jaime García-Mena[1]

[1] Departamento de Genética y Biología Molecular, Centro de Investigación y de Estudios Avanzados del Instituto Politécnico Nacional, Mexico City, CDMX, Mexico
[2] Departamento de Farmacología, Centro de Investigación y de Estudios Avanzados del Instituto Politécnico Nacional, Mexico City, CDMX, Mexico
[3] Hospital General ''Dr. José María Rodríguez'', Instituto de Salud del Estado de México, Ecatepec de Morelos, Estado de Mexico, Mexico

[*] These authors contributed equally to this work.

## ABSTRACT

**Background**. Human milk microbiota plays a role in the bacterial colonization of the neonatal gut, which has important consequences in the health and development of the newborn. However, there are few studies about the vertical transfer of bacteria from mother to infant in Latin American populations.

**Methods**. We performed a cross-sectional study characterizing the bacterial diversity of 67 human milk-neonatal stool pairs by high-throughput sequencing of V3-16S rDNA libraries, to assess the effect of the human milk microbiota on the bacterial composition of the neonate's gut at early days.

**Results**. Human milk showed higher microbial diversity as compared to the neonatal stool. Members of the Staphylococcaceae and Sphingomonadaceae families were more prevalent in human milk, whereas the Pseudomonadaceae family, *Clostridium* and *Bifidobacterium* genera were in the neonatal stool. The delivery mode showed association with the neonatal gut microbiota diversity, but not with the human milk microbiota diversity; for instance, neonates born by C-section showed greater richness and diversity in stool microbiota than those born vaginally. We found 25 bacterial taxa shared by both ecosystems and 67.7% of bacteria found in neonate stool were predicted to originate from human milk. This study contributes to the knowledge of human milk and neonatal stool microbiota in healthy Mexican population and supports the idea of vertical mother-neonate transmission through exclusive breastfeeding.

# INTRODUCTION

Human milk provides essential nutrients, bioactive substrates (*Fernández et al., 2013*), as well as prebiotics like the human milk oligosaccharides (HMOs) (*Ward et al., 2006*),

Corresponding author
Jaime García-Mena,
jgmena@cinvestav.mx

required for growth and development of the newborn and infant at least during the first six months of life. The human milk also contains a rich community of bacteria that has been proposed to originate from the neonate's oral cavity, the breast surface (mother's skin), the lobules and lactiferous ducts in the lactating women (commonly called ''Breastfeeding-Associated Microbiota''), or through an entero-mammary pathway. This last hypothesis states that maternal gut bacteria are translocated by dendritic cells through the intestinal epithelial barrier and are transported to the mammary glands via lymphatic circulation. From there, these bacteria colonize the gut of the breastfed neonate (*Sakwinska et al., 2016*; *LaTuga, Stuebe & Seed, 2014*; *Fitzstevens et al., 2017*; *Donnet-Hughes et al., 2010*; *Fernández et al., 2013*; *Jost et al., 2014*). Most reported studies are of microbiota of milk collected applying sanitization procedures to the mammary gland, and only few have studied the bacteria communities of the entire human milk collected without sanitization (*Ward et al., 2013*; *Sakwinska et al., 2016*).

The bacterial communities identified in the milk of healthy women are highly diverse and complex. Despite the great interindividual variability, several taxa have been identified as common constituents of milk microbiota, including *Staphylococcuss* spp., *Streptococcuss* spp., *Pseudomonas* spp., *Propionibacterium* sp., and *Lactobacillus* spp., as well as obligate anaerobic bacteria such as *Bifidobacterium* spp., *Clostridium* spp., and *Bacteroides* spp. (*Hunt et al., 2011*; *Jost et al., 2014*; *Fitzstevens et al., 2017*). These bacteria also represent the main groups involved in early gut colonization in healthy newborns (*Nagpal et al., 2017*).

Over the last few years, the vertical transmission of bacteria from the human milk to the infant gut has gained great interest as an important source or inoculum for bacterial colonization (*Grönlund et al., 2007*; *Nagpal et al., 2017*; *Simpson et al., 2018*). It is also proposed that gut microbiota acquisition begins *in utero* (*Aagaard et al., 2014*; *Collado et al., 2016*; *Parnell et al., 2017*; *Stinson et al., 2019*), being it another source of bacteria for the neonate. Likewise, although there is evidence of translocation of specific bacterial taxa from milk to the infant gut, the proportion of microbiota from the human milk that contributes to the colonization during the first days after birth has not been extensively characterized (*Pannaraj et al., 2017*).

The human milk microbial composition is influenced by factors that impact the early development of the gastrointestinal microbiota in the neonate, which include mode of delivery, diet, lifestyle, and geographical area where the mother lived during pregnancy (*Khodayar-Pardo et al., 2014*; *Cabrera-Rubio et al., 2016*; *Kumar et al., 2016*; *Mueller et al., 2017*). In Mexico, there are no published studies examining the bacterial ecosystems of milk and/or neonate gut in a cohort. Most research done to date, has focused on Spanish (*Gomez-Gallego et al., 2016*), Irish (*Murphy et al., 2017*), Canadian (*Moossavi et al., 2019*), western Italian (*Biagi et al., 2017*), and United States Caucasian populations (*Hunt et al., 2011*), whose ethnicities, lifestyles and environment exposures differ from the Mexican population. We believe that in the Mexican population the neonate's intestinal microbiota diversity is determined mostly by the microbiota found in the milk of its mother, and the delivery mode. In this context, the aim of this cross-sectional and comparative study was to evaluate the association of the human milk bacteria and the delivery mode with the neonate gut bacterial composition in a cohort of Mexican population based on the sharing

of milk/ neonate gut bacteria. We also describe the predicted metabolic pathways in these bacterial communities.

## MATERIALS & METHODS

### Study design and selection of subjects

This descriptive cross-sectional study included 67 mother-neonate pairs recruited at General Hospital ''Dr. José María Rodríguez'', located in Ecatepec–de–Morelos, State of Mexico (19°36 ′35″ N, 99° 3′36″W), between November 2017 and January 2018. Donors were healthy lactating women, and their healthy, full-term exclusively breastfed neonates. Milk and stool samples were collected from each mother–neonate pair between 1–6 days postpartum. Inclusion criteria to select the pairs were: (1) Mexican origin for at least two generations, (2) gestational age between 37 and 41 weeks, (3) spontaneous vaginal delivery or non-elective C-section, (4) birth weight greater than 2,500 g and less than 4,500 g, and (5) Apgar score greater than 7 at 5 min after birth. Exclusion criteria: (6) probiotics and alcohol consumption, (7) smoking, (8) diabetes, overweight and obesity before or during pregnancy, and (9) antibiotics use during the last trimester of pregnancy and prior to sample collection. Based on a questionnaire, sociodemographic and clinical information were recorded (maternal age, gestational age at delivery, delivery mode, sex, and age of newborn). Written informed consent was obtained from all donors before starting the study in accordance with the Declaration of Helsinki 2013. The protocol was approved by the Ethics Committee of the General Hospital ''Dr. José María Rodríguez'' (Project identification code: 217B560002018006).

### Sample collection

All samples were obtained by one member of the research team wearing sterile gloves. Each milk-neonatal stool sample pair was collected the same day in the morning up to 2 h after the neonate was breastfed. Milk for the study was manually collected (5–10 mL) into a sterile polypropylene tube without breast sanitization to give a more representative analysis of the bacteria ingested by the suckling neonate. At the same time, fecal samples were taken after 20 min at most, directly from diapers into sterile containers with the help of sterile tongue depressors. All samples were immediately transported to the laboratory using cold packs and dispensed in aliquots of one mL of milk or 200 mg of stool and stored at −20 °C until processing for DNA extraction within 24 h of receipt.

### DNA extraction

Prior to DNA extraction, one mL of milk was centrifuged at 10,000 g, 15 min at 4 °C in a refrigerated centrifuge (Eppendorf 5415R) and fat was removed using a sterile dental cotton roll. Aqueous supernatant was removed by decantation, the pellet resuspended in 1.0 mL sterile PBS pH 7.4, then recentrifuged at 10,000 g for 15 min. The obtained pellet was resuspended in 300 μL of PBS pH 7.4 and processed for DNA extraction using FavorPrep Milk Bacterial DNA Extraction Kit (Cat.: FAMBD001, Favorgen, Biotech Corp, Taiwan) following the manufacturer's instructions. Fecal DNA was extracted from 200 mg stool samples using a QIAamp DNA Stool Mini Kit (Cat.: 12830-50, Qiagen, Netherlands),

following the manufacturer's instructions. In both cases 300 µL of PBS pH 7.4 was used as negative control for DNA extraction. The DNA concentration in samples was measured at 260/280 absorbance using a Nano Drop 2000 spectrophotometer (Thermo Scientific, USA), no absorbance was detected for the negative controls. The DNA integrity was evaluated by electrophoretic fractionation in 0.5% agarose gel. DNA was stored at −20 °C until library preparation and sequencing.

## Preparation of the 16S rDNA library and high-throughput sequencing

For each DNA sample, a ∼281 bp amplicon containing the V3 hypervariable region of the 16S RNA gene was amplified using V3-341F forward primer (set of barcodes 1–100) complementary to positions 340–356 of the *Escherichia coli* 16S rDNA molecule *rrnB* GenBank J01859.1, and the V3–518R reverse primer complementary to positions 517–533 of same molecule (*Fierer et al., 2008*; *Murugesan et al., 2015*) (Table S1). All PCR reactions were performed in a final volume of 25 µL 1X SYBR Green PCR Master Mix (Bio-Rad Laboratories Cat# 1725270), 0.3 µM of each primer, and 10–25 ng of each DNA template. The PCR conditions were as previously reported with exception that 30-cycles were used (*Murugesan et al., 2015*). The ∼281 bp amplicon was not observed for the negative controls thus they were not sequenced. For library preparation equal mass amounts of each 1–100 barcoded amplicons were quantified by gel densitometry and pooled. The mixture was purified using E-Gel iBase Power System (Invitrogen). The libraries size and concentration were confirmed using the Agilent 2100 Bioanalyzer system and High Sensitivity DNA Kit (Agilent, USA). High-throughput sequencing was performed using Ion OneTouch 2, Ion PGM Template OT2 200 Kit v2 DL (Life Technologies, California, USA), Ion 318 Chip Kit v2 and Ion Torrent PGM System as previously described (*Chávez-Carbajal et al., 2019*). After sequencing, reads were filtered by the PGM software to exclude low quality and polyclonal sequences. The quality control of sequences was performed using (*FastQC, 2019*) (https://www.bioinformatics.babraham.ac.uk/projects/fastqc/), and all reads were trimmed to 200 nt length using Trimmomatic v0.36. Filtered and demultiplexed FASTQ files were converted into FASTA files, concatenated into a single file, and then processed with multiple QIIME (Quantitative Insights into Microbial Ecology) v1.9.0 scripts (*Caporaso et al., 2010*). DNA sequences were classified into Operational Taxonomic Units (OTUs) using closed based picking parameters with a 97% similarity level against Greengenes database v13.8. The sequence and corresponding mapping files for all samples used in this study were deposited in the NCBI BioSample repository (accession number: PRJNA548324).

## Microbial abundance and diversity analyses

The relative abundance of bacterial communities at phylum and family taxonomic levels was determined for human milk or neonatal stool samples using QIIME pipeline v1.9.0. The linear discriminant analysis effect size program (LEfSe v1.0) was used to detect significant differences in the relative abundances of bacterial taxa among milk and stool samples, and vaginal and C-section delivery mode samples. We used the LDA (linear discriminant analysis) to estimate the effect size of each taxa between groups with LDA–scores ≥ 2.5 (*Segata et al., 2011*). To characterize the microbial diversity patterns, we calculated alpha

and beta diversities. Prior to calculate alpha diversity, the OTU table was rarefied at 10,000 sequences per sample (samples with <10,000 were omitted) using a "single_rarefaction.py" QIIME script for the four alpha diversity metrics: observed species (number of unique OTUs), Chao1 index (bacterial richness estimator), and the community diversity Simpson (dominance) and Shannon (evenness) indexes were determined using phyloseq and ggplot2 packages in R environment (v3.4.4). The Effect-size was measured using the Hedges' g statistic and calculated with STATA SE 10.1 software. For beta diversity, the dissimilarity was estimated using weighted and unweighted UniFrac analyses. A two-dimensional scatter plot was generated using principal coordinate analysis (PCoA) with QIIME.

## Shared OTUs, core microbiota and microbial source tracking analysis

To determine the number of OTUs in the microbial community shared between the human milk and the neonatal stool samples the shared_phylotypes.py QIIME script was run, then a Venn Diagram was generated using the Bioinformatics and Evolutionary Genomics web tool (*Shade & Handelsman, 2012*). Next, we used the compute_core_microbiome.py QIIME script to identify which taxa are shared in at least 50% of mother-neonate pairs, and a heatmap of counts was made in R environment (gplots and RColorBrewer packages). We performed Source Tracker analysis to predict the origin of OTUs in each neonatal stool sample using the corresponding human milk as potential source, to estimate the proportion of bacteria present in the neonatal stool attributable to the human milk. This analysis was made in QIIME platform using Source Tracker (v0.9.5) software (*Knights et al., 2011*). A file with the mean of all these data was used to show the proportion of OTUs present in the neonatal stools corresponding to "human milk" or "unknown source" which indicate other possible sources not evaluated. Data were visualized as pie chart plots for each sample and for total samples.

## Metagenome prediction with PICRUSt

We used Phylogenetics Investigation of Communities by Reconstruction of Unobserved States (PICRUSt, v1.1.1) (*Langille et al., 2013*) to predict the metabolic function of the metagenomes from 16S rRNA gene data set, with Kyoto Encyclopedia of Genes and Genomes (KEGG) orthologs classification database at hierarchy level 3 pathways. Statistical Analysis of Taxonomic and Function software (STAMP v2.1.3) was used to determine significant differences in abundance of OTUs and metabolic pathways.

## Statistical methods

The epidemiological data were reported as mean $\pm$ standard deviation (SD), or frequencies and percentages. Two-tailed student's $t$-test, Mann-Whitney U, or Wilcoxon signed–rank nonparametric test were assessed to compare groups using SPSS v23.0 software (SPSS, Inc). ANOSIM and Adonis were used for category comparisons of phylogenetic distance matrices (UniFrac). Linear regression was used to know the relationship between microbiota diversity as the dependent variable and maternal and neonatal age included as covariates; $p < 0.05$ was considered statistically significant. The Benjamini $-$Hochberg

(BH) correction method was used to estimate the false discovery rate (FDR) and filter the data where a $q$-value <0.05 was considered statistically significant.

## RESULTS

### Participating women were from a poor urban polluted area

We characterized the human milk microbiota of 67 lactating relatively healthy Mexican women aged 14 to 41 years-old, and the gut microbiota present in the stool of their respective neonates aged <6 days-old, fed exclusively with human milk. In this cohort, most women were housewives living in "Ecatepec–de–Morelos" (19°29′4.56″ N, 99°7′6.96″W), an overcrowded municipality area (8,860 inhabitants/km$^2$) of the "Estado–de–México" a state located in the central part of Mexico (*INEGI, 2019*). At 2,248 m over sea level, "Ecatepec–de–Morelos" has a subtropical highland climate (Köppen: Cwb), and 75.2% of the population lives in a situation of moderate to extreme poverty (*CONEVAL, 2019*). Participating women had mostly a high school or college educational level. With respect of neonates, most of them were females born by vaginal delivery (Table 1).

### Proteobacteria was the most abundant phylum in the mother-neonate pair

We characterized the bacterial diversity in the human milk and stool samples collected from the mother–neonate pairs by high-throughput DNA semiconductor sequencing of V3-16S rDNA libraries. In general, we obtained 9,575,537 raw sequence reads with a mean length of 169.3 bp (±46.1) for the total of 134 samples analyzed; 4,240,314 for human milk and 5,335,223 for neonatal stool with a phred33 average value of 31. A mean length 170 bases were selected for analyses (Fig. S1). The alpha rarefaction curves show that the deep of sequencing among samples was sufficient to process the data (Table S2).

We found that Proteobacteria, Firmicutes, Actinobacteria, and Bacteroidetes accounted for 97.19% of sequences in human milk (Fig. 1A) and 98.03% in neonatal stool (Fig. 1B). The phylum Proteobacteria was more abundant in human milk (55.40% ± 32.1) than in neonatal stool (36.70% ± 31.0) being this the only phylum with a statistical significance difference ($p = 0.001$, $q = 0.041$). Although with no statistically significant difference, the relative abundance of Firmicutes was larger in neonatal stool (32.10% ± 33.2) than human milk (25.80% ± 28.9) ($p = 0.243$, $p = 1.00$); the phylum Actinobacteria was more abundant in neonatal stool (18.73% ± 23.5) than human milk (13.20% ± 11.7) ($p = 0.088$, $q = 1.00$); while the relative abundance of Bacteroidetes was higher in neonatal stool (10.50% ± 21.5) than in human milk (2.79% ± 9.9) ($p = 0.009$, $q = 0.185$). In addition, the phyla Acidobacteria, Cyanobacteria, Fusobacteria, Chloroflexi and Armatimonadetes with less than 1% of relative abundance, were grouped as "Others", and they accounted for 2.83% (± 1.12) in human milk and 1.94% (± 0.47) in neonatal stool. These differences had no statistical significance ($p = 0.598$, $q = 1.00$) (Figs. 1A, 1B).

In addition to the phylum, we analyzed the bacterial composition at family level in both groups (Fig. 1C and Table S3). We found 14 predominant families with a relative abundance ≥ 1% in at least 97% of samples. The relative abundance of Staphylococcaceae, Sphingomonadaceae, Rhodobacteraceae, Bradyrhizobiaceae, and Propionibacteriaceae

**Table 1  Sociodemographic and clinical characteristics of the study population.**

| Maternal Data | n (%) |
| --- | --- |
| Years of age[a] | 22.12 ± 5.7 |
| Birthplace | |
| State–of–Mexico | 41 (61.2) |
| Mexico City | 14 (20.9) |
| Other (Oaxaca, Puebla, and Veracruz states) | 12 (17.9) |
| Main activity | |
| Housewife | 60 (89.6) |
| Student | 1 (1.49) |
| General employee | 6 (8.96) |
| Educational level[b] | |
| Elementary school | 17 (25.4) |
| High school | 21 (31.3) |
| College | 25 (37.3) |
| None | 4 (5.97) |
| Parity | |
| Uniparous | 27 (40.3) |
| Multiparous | 40 (59.7) |
| Neonate's delivery mode | |
| Vaginal | 46 (68.7) |
| C-Section | 21 (31.3) |
| **Neonatal data** | **n (%)** |
| Age at sample collection, days[c] | |
| <3 | 61 (91.0) |
| 4–6 | 6 (9.0) |
| Sex | |
| Female | 40 (59.7) |
| Vaginal | 28 (41.8) |
| C-section | 12 (17.9) |
| Male | 27 (40.3) |
| Vaginal | 18 (26.9) |
| C-section | 9 (13.4) |

**Notes.**
[a] Expressed as mean ± standard deviation, n—sample number, State–of–Mexico (19.6049°N 99.0606°O), Mexico City (19.4285° N 99.1277°O), Other states: Oaxaca (17.0654°N 96.7236°O), Puebla (19.0379°N 98.2035°O), Veracruz (19.181°N 96.1429°O).
[b] Equivalent based on U.S. Department of Education (*McFarland et al., 2018*).
[c] Postpartum days.

was higher in human milk with respect to neonatal stool. In contrast, Pseudomonadaceae, Clostridiaceae, and Bifidobactericeae showed higher relative abundance in neonatal stool. On the other hand, Streptococcaceae, Weeksellaceae, and Lachnospiraceae were equally abundant in both groups. The relative abundance of some of these families showed higher inter-individual variation among members in each group (Table S3).

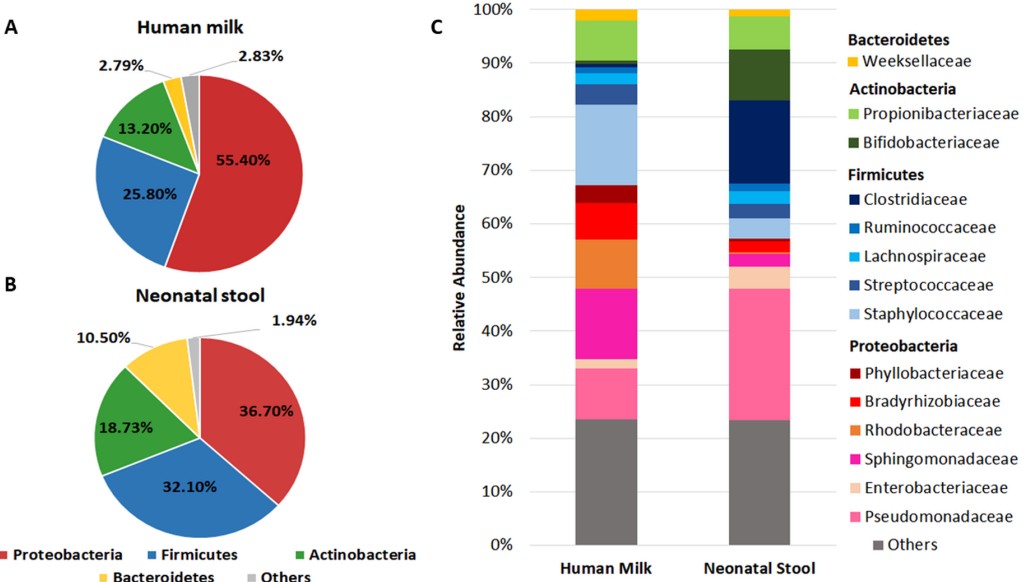

**Figure 1 Relative abundance of predominant bacterial taxa (phyla and families) in human milk and neonatal stool.** Abundance of each phylum in human milk (A) and neonatal stool (B). Comparison between groups was calculated using parametric $t$-test for paired samples followed by BH correction: Proteobacteria ($p = 0.001$, $q = 0.041$); Firmicutes ($p = 0.243$, $q = 1.00$); Actinobacteria ($p = 0.088$, $q = 1.00$); Bacteroidetes ($p = 0.009$, $q = 0.185$); Others ($p = 0.598$, $q = 1.00$). (C) Relative abundance of dominant bacterial families for each group.

## There is a significant difference in the abundance of some bacterial taxa between human milk and neonate stool

We used LEfSe analysis to identify differences in the abundance of bacterial taxa between human milk and neonatal stool microbiota, using an LDA score cutoff of 3.5. Fourteen taxa were predominant in the human milk samples and six taxa in neonatal stools. Identified taxa in human milk included ten genera *Staphylococcus, Kaistobacter, Paracoccus, Pseudomonas, Bradyrhizobium, Methylobacterium, Acinetobacter, Propionibacterium, Corynebacterium,* and *Microbacterium*; three families Phyllobacteriaceae, Sphingomonadaceae, Gemellaceae, and the order Streptophyta (average $p = 1.91E-5$ and $q = 1.07E-3$). Likewise, in the stool samples, we found members of three families Pseudomonadaceae, Clostridiaceae, and Enterobacteriaceae; and three genera *Bifidobacterium, Clostridium,* and *Bacteroides* (average $p = 4.26E-5$ and $q = 9.67E-5$) (Fig. 2 and Table S4).

## The Human milk microbiota diversity was higher than the diversity in neonatal stool

We estimated the alpha diversity of the microbiota in human milk and neonatal stool samples and Mann–Whiney U test to find significant differences between both groups (Figs. 3A–3D and Table S5). To estimate microbial richness, we used the Chao1 index (Effect size 0.894, $p < 0.001$) and Observed number of species (Effect size 0.861, $p < 0.001$) which revealed a difference between both groups showing higher richness in milk microbiota. With respect to the diversity and dominance, the Shannon (Effect size 0.745, $p < 0.001$) and

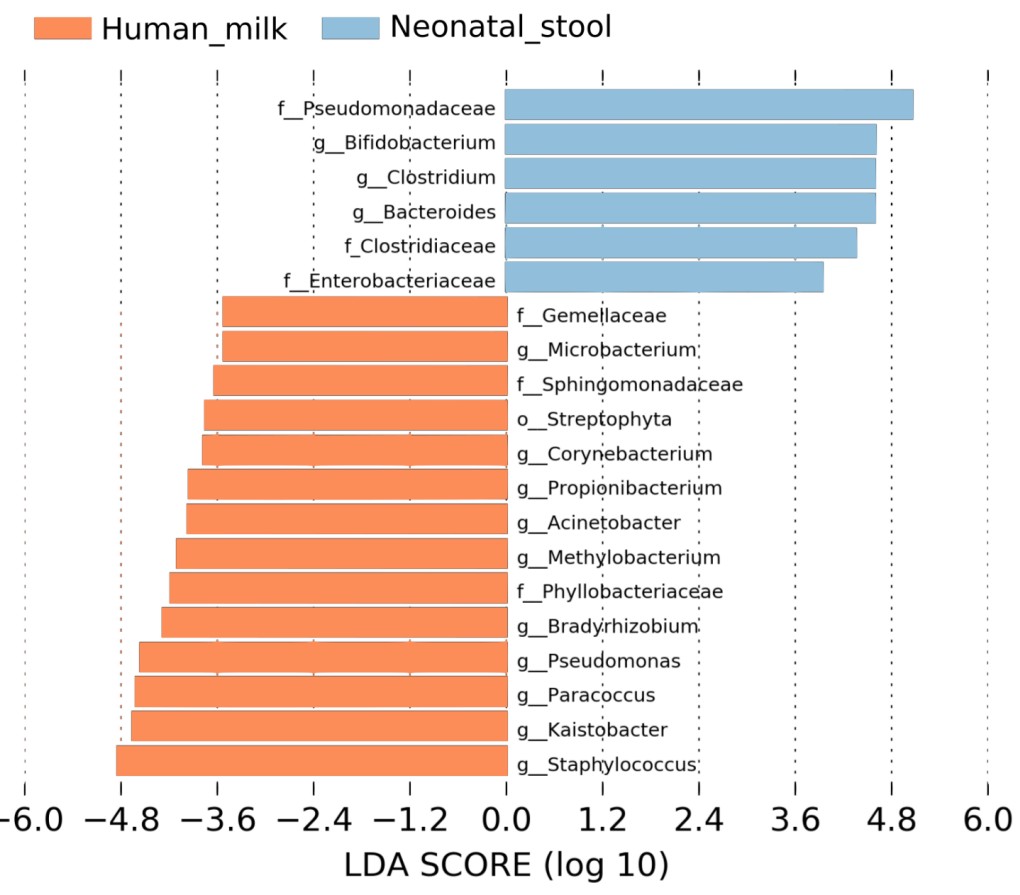

**Figure 2** **Linear discriminant analysis (LDA) effect size (LEfSe) comparison of differentially abundant bacterial taxa between human milk and neonatal stool.** Horizontal bars represent the effect size for each taxon: light blue color indicates taxa enriched in neonatal stool group, and crusta color indicates taxa enriched in milk group. LDA score cutoff of 3.5 was used to discriminate bacterial taxon. Statistically significant values are shown in Table S4.

Simpson (Effect size 0.584, $p = 0.006$) indexes showed higher bacterial diversity in human milk samples compared to the neonatal stool microbiota. Only the Simpson index did not show significant difference after the FDR correction (Table S5). Next, the beta diversity based on unweighted UniFrac analysis revealed two clusters separating both groups of samples as is illustrated by the three 2D images of the analysis (ANOSIM, $R = 0.289$, $p = 0.001$) (Figs. 3E –3G). To confirm these statistical differences between both bacterial communities Adonis test was calculated ($R^2 = 0.949$, $p = 0.001$) and Weighed UniFrac analysis was also performed (Fig. S2).

## The delivery mode is associated with the neonatal gut microbiota but not the human milk microbiota diversity

We observed that the bacterial diversity and richness of the neonatal fecal microbiota are associated with the delivery mode, nevertheless a similar effect was not observed for the human milk microbiota (Table S6). The observed species (Effect size 0.071, $p = 0.003$),

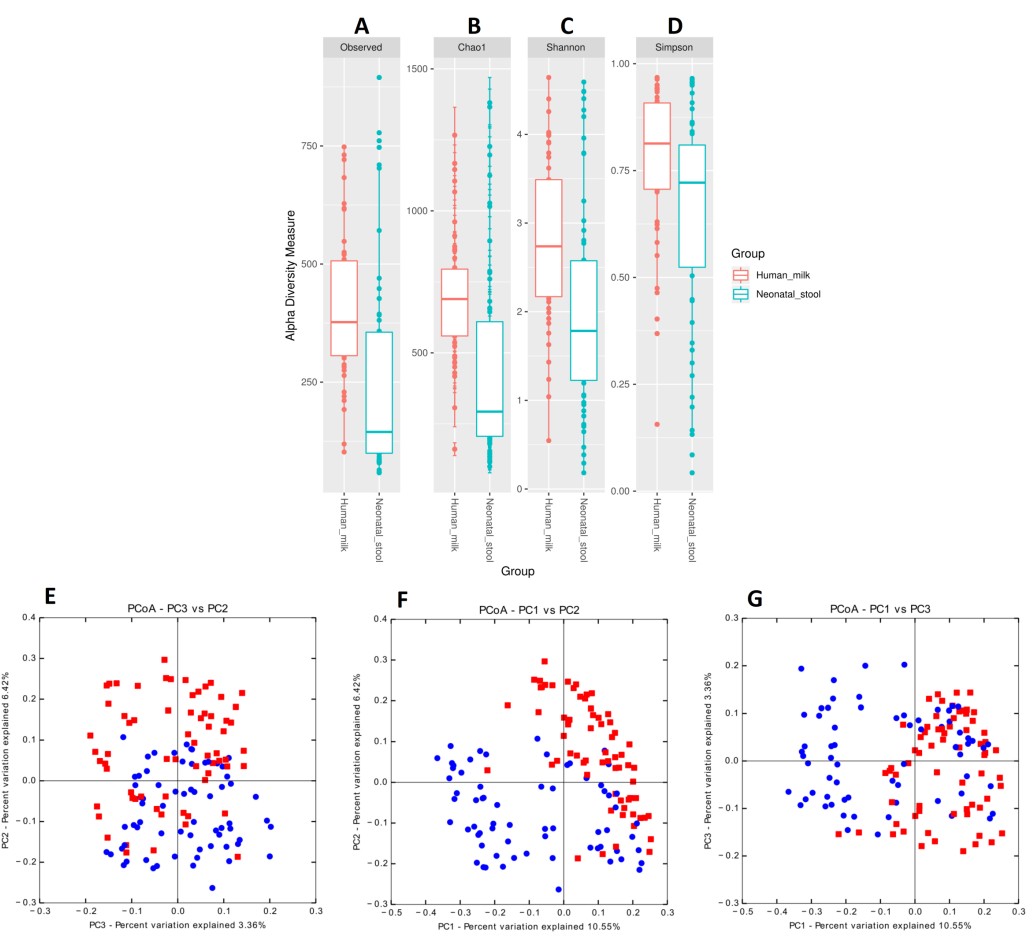

**Figure 3 Bacterial diversity of human milk and neonatal stool microbiota.** Alpha diversity based on (A) observed number species ($p < 0.001$), (B) Chao1 ($p < 0.001$), (C) Shannon ($p < 0.001$) and (D) Simpson ($p = 0.006$) indexes. Mann–Whitney $U$-test was used to find significant differences. Beta diversity analysis. Two-dimensional scatter plots were generate using PCoA based on unweighted UniFrac distance metric. (E) PC3 vs PC2, (F) PC1 vs PC2, and (G) PC1 vs PC3. Both groups significantly differed according to ANOSIM similarity test ($R = 0.289$, $p = 0.001$) and Adonis statistical test ($R^2 = 0.949$, $p = 0.001$). Human milk samples ($n = 67$) are plotted as red dots and neonatal stool ($n = 67$) as blue dots. Statistically significant values are in Table S5.

Chao1 richness indexes (Effect size 0.067, $p < 0.006$) and Shannon diversity indexes (Effect size 0.086, $p = 0.046$) were significantly higher in stool samples from neonates delivered by C-section compared to those born by vaginal delivery (Figs. 4A –4D and Table S6). Likewise, the beta diversity analysis revealed that the fecal microbiota of C-section delivered neonates were different to those born vaginally (ANOSIM: $R = 12.52$, $p = 0.006$; Adonis: $R^2 = 0.0401$, $p = 0.001$) (Fig. S3).

We also used LEfSe analysis to identify bacteria with relative abundance, significantly different between groups. We found a significant larger number of taxa (thirteen genera and three families) in the stool of neonates born by C-section compared with those born by vaginal delivery (three genera, two families and one order). Among these, *Staphylococcus*
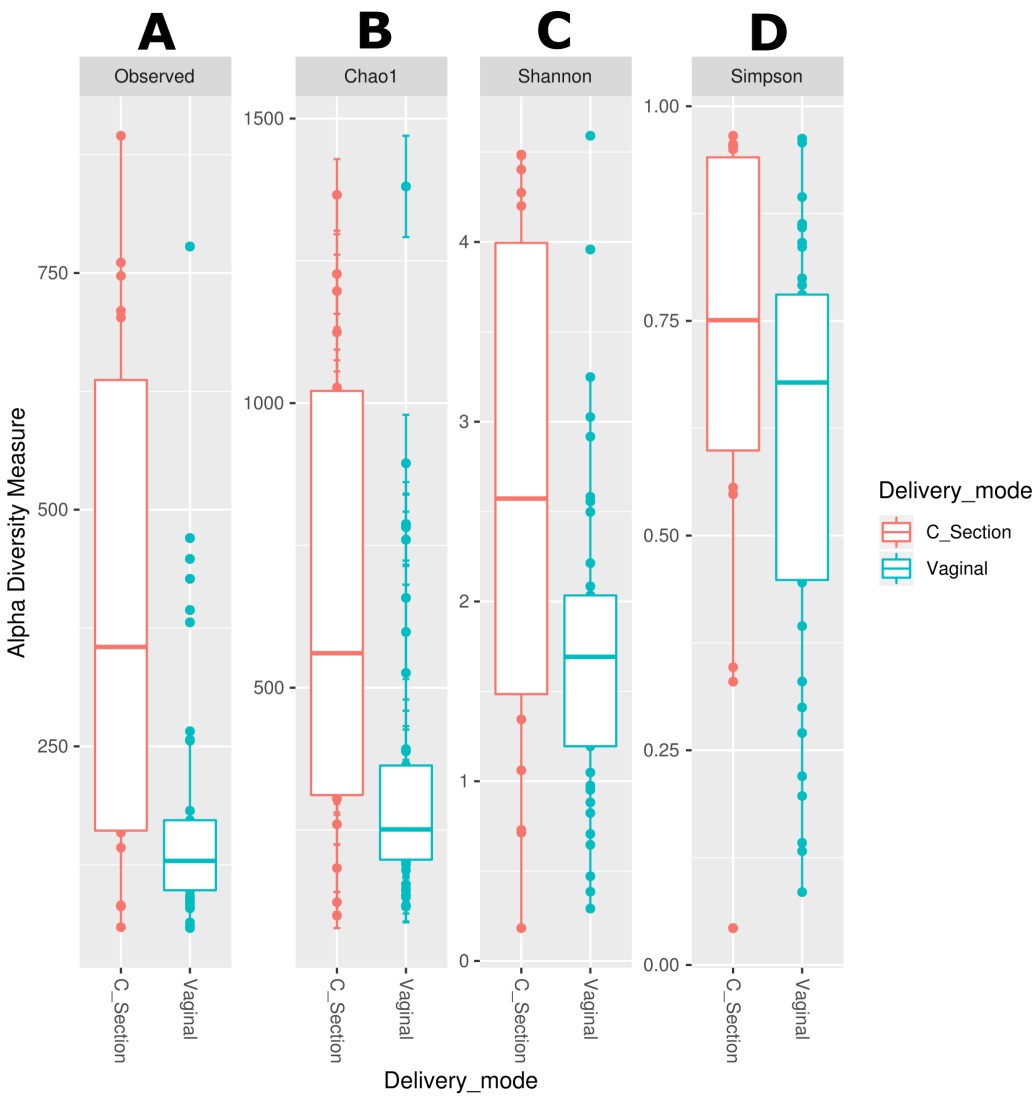

**Figure 4 Alpha diversity in neonatal stool samples from neonates born by C-section (n=19) or vaginal delivery mode ($n = 41$).** (A) Observed number species ($p = 0.003$), (B) Chao1 ($p = 0.006$), (C) Shannon ($p = 0.046$), and (D) Simpson ($p = 0.082$) indexes. The diversity indexes were calculated using Mann−Whitney $U$ test where $p < 0.05$ was considered significant (Table S6). Labels beside the graphics indicates the delivery mode.

was the most abundant genus in the neonates delivered by C-section, and the family Pseudomonadaceae was the most abundant in neonates born by vaginal delivery (Fig. S4). Finally, we evaluated the association of maternal and neonatal age on the bacterial diversity applying a lineal regression analysis. For the human milk the richness (Chao1) (Fig. S5A) and the diversity (Shannon) (Fig. S5B) have a slight tendency to decrease as the age of the mother increases; same indexes slightly increased with the days after delivery for the neonatal stool (Figs. S5E, S5F), and for the human milk, the richness does not have an

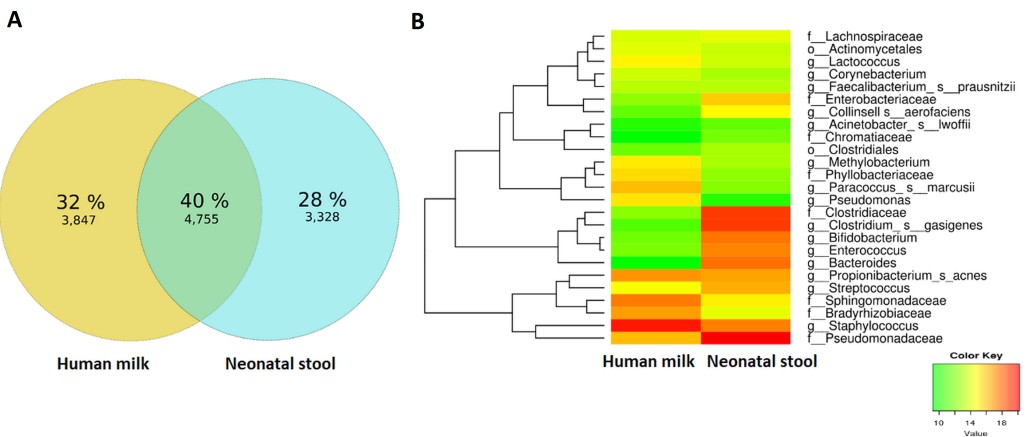

**Figure 5** **Analyses of shared OTUs in human milk/ neonatal stool and its abundance.** (A) Venn diagram showing unique and shared OTUs between human milk and neonatal stool samples. (B) Heatmap shows shared OTUs counts between taxa of human milk and neonatal stool groups. Included counts were present in at least 50% of paired samples and calculated by the compute_core_microbiome.py Qiime script. Color Key from green to red indicates increasing absolute abundance in natural logarithm of counts. Green color indicates lowest abundance while red color highest abundance, with taxa along the $Y$-axis and samples along $X$-axis.

apparent change (Fig. S5C) while the diversity has a tendency to increase with the days after delivery (Fig. S5D).

## Specific bacterial taxa are transferred from human milk to the neonatal gut from the first day of life

As a first approach to explore the transferring of bacteria from human milk to the neonatal gut, we focused our attention to OTUs shared between each homologous mother—neonate pair. In general, from the 11,930 observed OTUs only 4,755 (39.85%) were shared. We found that 3,847 OTUs (32.24%) from human milk were not found in the neonate stool, while 3,328 OTUs (27.89%) from neonate stool were not found in the human milk (Fig. 5A). Relevant shared taxa between human milk and neonatal stool were the families Pseudomonadaceae, Clostridiaceae, Sphingomonadaceae, and Bradyrhizobiaceae, and the genera *Clostridium gasigenes*, *Bacteroides* spp., *Bifidobacterium* spp., *Staphylococcus* spp., *Enterococcus* spp., *Streptococcus* spp., and *Propionibacterium acnes* (Fig. 5B and Table S7).

Next, we used microbial source tracking analysis to estimate the proportion of bacteria in the neonatal stool which originated from the human milk (Fig. S6). Data showed that neonates received 67.8% (±36.5) of bacteria from human milk, while the remaining 32.2% (±36.5) came from unknown sources ($p < 0.006$, Wilcoxon signed–rank test) (Fig. 6A). The taxonomic analysis of the bacteria in the fecal samples identified as "Unknown Source Group" showed a high significant relative abundance of the orders Clostridiales followed by Bacteroidales, Lactobacillales, and Enterobacteriales, which was different to the abundance found in "Human Milk Group" (Fig. 6B).

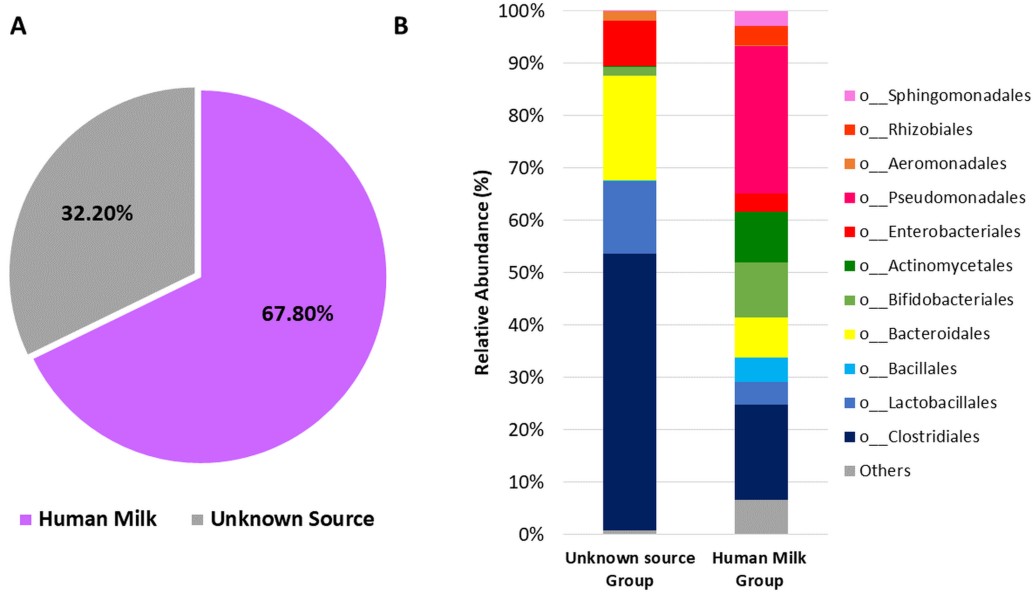

**Figure 6** **Probable origin of bacteria in the neonatal stool.** (A) Microbial source tracker analysis showing the proportion of bacteria identified in the neonatal stool classified by source ($p < 0.001$, Wilcoxon signed-rank test). (B) Relative abundance of most common bacterial orders found in the neonatal stool classified by Qiime source tracker analysis as ''Human milk'' and ''unknown source''.

## The predicted functional metagenome shows greater carbohydrate metabolism in neonate gut and greater lipid metabolism in human milk microbiota

We determined the functional metabolic pathways present in the human milk and neonatal stool microbiota by PICRUSt analysis using the OTU table. At level three of analysis (the most specific) 10 KEGG pathways out of 92, showed statistically significant difference between human milk and neonate stool microbiota (average $p = 3.80E{-}03$, average $q = 8.74E{-}03$) (Fig. 7 and Table S8). The functional pathways were related to energy metabolism, bacterial colonization, and immune function. The human milk showed a high abundance of bacterial metabolic pathways associated to fatty acid metabolism, whereas the fecal microbiota from neonates had a higher abundance of pathways involved in carbohydrate metabolism, vitamin B6 metabolism, bacterial colonization, and immune function.

## DISCUSSION

Human milk provides all required nutrients for infant nourishment; in addition, it contains a community of bacteria transferred through breastfeeding that plays a fundamental role in the development of the infant gut microbiota. In this cross-sectional study, we characterized the profile of the human milk microbiota from healthy Mexican mothers and the fecal microbiota of their neonates, finding that human milk contributes with the 67.7% of the bacteria within the first six days postpartum. Previous studies have reported both similarities and differences when comparing bacterial profiles of paired samples (human

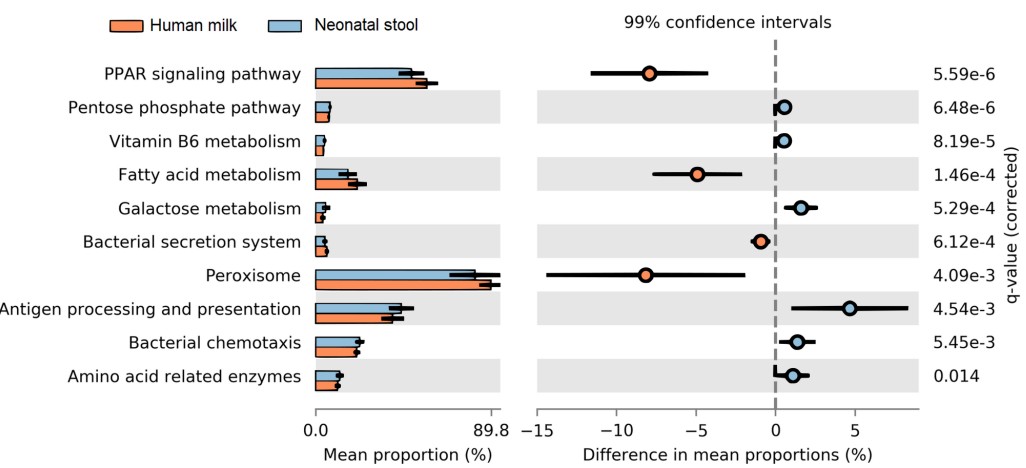

**Figure 7** **Prediction of functional microbial metabolic pathways using PICRUSt analysis (KEGG level three).** The abundance of 10 statistically significant metabolic pathways between human milk (crusta color) and neonatal stool (light blue color) bacterial communities. All statistically significant pathways ($q < 0.05$) are included in Table S8.

milk/neonatal stool) in different populations (Table 2). In our work we found high levels of Proteobacteria (55.4%) and Firmicutes (25.8%), represented by members of the Pseudomonadaceae and Staphylococcaceae families, present in 71.6% and 82.1% of milk samples respectively (Table S3).

Pseudomonadaceae family, for example, has been reported in human milk with an abundance of up to 61% during the first 30–days postpartum in a Canadian population (*Ward et al., 2013*), and an abundance of 17% during the 6 weeks postpartum in an Irish population (*Murphy et al., 2017*), while the Staphylococcaceae family has been reported with an average abundance of 20% in several reports (Table 2). We believe that the presence of Staphylococcaceae is a common feature of human milk microbiota, regardless of the geographical location. Conversely, the Sphingomonadaceae, and Rhodobacteraceae families represented by the *Kaistobacter* and *Paracoccus* genera respectively, were abundant in human milk samples in our study, nevertheless, these two taxa have been poorly reported in other studies on human milk microbiota in healthy women (Table 2). To our knowledge, there is no available information about the possible functional role of these bacteria in human milk on regard of the neonate gut, but the Sphingomonadaceae family, has been also detected in high abundance in nipple skin and nipple aspirate fluid samples in healthy women (*Chan et al., 2016*; *Hieken et al., 2016*). It is suggested this family contributes to the maintenance of healthy breast tissue and protects against breast cancer through stimulation of the host immune cells (*Chan et al., 2016*) (Table 3). A relevant finding was the low relative abundance in our milk samples of Bifidobacteriaceae (0.73%), Streptococcaceae (3.73%) families and genus *Lactobacillus* (0.21%); since these taxa have been reported in high abundance as part of a predominant "core microbiota" in human milk from healthy women in other studies (*Fernández et al., 2013*; *Jiménez et al., 2015*; *Kumar et al., 2016*). For example, in the CHAMACOS study performed in Mexican American women suffering

of overweight and obesity (*Davé et al., 2016*), and in Peruvian healthy women (*Lackey et al., 2019*), *Streptococcus* was the dominant genus (73.8% and 49.6% respectively), however in our study, this taxa barely reached a relative abundance of 2.3%. We think that for human milk there is a "core microbiota" plus additional resident bacteria which are unique for each population, whose variability is influenced by the genetic composition, geographical area, lactation stage, diet, lifestyle, and environmental conditions that surround the mother.

The characterization of the neonate fecal microbiota showed less diversity than the human milk microbiota, as well as a high dominance of bacterial groups (Fig. 3), which suggest that in the first six days of life, only few selected taxa contribute to the initial colonization of the gut microbiota. The high dominance of Pseudomonadaceae (24.5%) in 77.6% of neonate fecal samples, suggests a possible selection process occurred in the infant gut (Fig. 5B). We think these bacteria come mostly from human milk and constitute an inoculum for the gut colonization at early age (Fig. 6B). It has been already described the presence of *Pseudomonas aeruginosa* in the gut of healthy newborns during the first week of life without any manifestation of disease (*Borderon et al., 1990*). Although the presence of *Pseudomonas* spp. in several body niches have been documented in healthy subjects (*Dekio et al., 2005*; *Urbaniak et al., 2014*; *Ozkan et al., 2019*), a potential functional role remains undefined. We can speculate these bacteria are selected for its ability to migrate and bind to the mucosal barrier of the epithelial cells (*Ruch & Engel, 2017*).

Several studies have reported the presence of common bacteria between human milk and infant stool. In fact, during the first month of life, breastfed infants share up to 28% of their fecal bacteria with their mother's milk bacteria as reported for Spanish, Canadian, Finnish, American and Italian populations (*Martín et al., 2012*; *Azad et al., 2013*; *Jost et al., 2014*; *Pannaraj et al., 2017*; *Ferretti et al., 2018*). In this study, despite differences in the bacterial diversity between the milk and stool samples, we observed the presence of 25 shared taxa in at least 50% of mother–neonate pairs, which can be an indicator of vertical transmission. These 25 taxa represent 40% of all detected bacterial OTUs (Fig. 5); such as *Staphylococcus*, Pseudomonadaceae, and *Propionibacterium acnes* (currently named *Cutibacterium acnes*, *Rocha Martin et al., 2019*). Previous studies have revealed that during breastfeeding period, *Staphylococcus* can reach the mammary gland from the gut (entero-mammary translocation) or the source can be the maternal skin (retrograde flow), constituting the most dominant bacteria in human milk (*Jiménez et al., 2008*; *Fernández et al., 2013*; *Urbaniak et al., 2014*; *Jiménez et al., 2015*; *Urbaniak et al., 2016*). Other species which are normally present on adult skin such as *Propionibacterium acnes*, *Streptococcus*, and *Corynebacterium* (*Jiménez et al., 2008*; *Grice & Segre, 2011*) were also abundant in human milk samples in our study. This is not surprising considering that skin bacteria have access to the mammary ducts through the nipple (*Ramsay et al., 2004*) and can spread within the mammary glands, independently of lactation (*Urbaniak et al., 2014*), probably for this reason, these bacteria have been also observed in human milk collected aseptically (*Hunt et al., 2011*). Some of breastfeeding related taxa mentioned above, such as the family Bifidobacteriaceae have been reported as pioneers of the infant gut due to its ability to metabolize human milk oligosaccharides (HMOs) to proliferate (*Yu et al., 2013*), in other cases the presence of the HMOs promote the growth of microorganisms

**Table 2  Selected studies of paired human milk–infant gut microbiota profiles in different populations.**

| Population | TS | n | D | Method | Predominant taxon human milk | Predominant taxon infant stool | Reference |
|---|---|---|---|---|---|---|---|
| Mexican | CS | 67 | 0–6 | Ion Torrent/(V3) | *Staphylococcus, Kaistobacter, Paracoccus, Pseudomonas.* | Pseudomonadaceae, *Bifidobacterium, Clostridium, Bacteroides.* | This study |
| Hispanic-Latino-White | LG | 90 | 1–7 | Illumina/(V3–V4) | Moraxellaceae, Staphylococcaceae, Streptococcaceae, Pseudomonadaceae | Bifidobacteriaceae, Enterobacteriaceae. | *Pannaraj et al. (2017)* |
| Peruvian | CS | 42 | 30–90 | Illumina/(V1–V3) | *Streptococcus, Staphylococcus, Rothia.* | *Escherichia/Shigella, Streptococcus, Veillonella, Enterococcus* | *Lackey et al. (2019)* |
| Spanish | CS | 20 | 7–90 | qPCR, culture | *Staphylococcus, Bifidobacterium, Lactobacillus.* | *S. epidermidis, Bifidobacterium brevis, B. longum, Lactobacillus casei, L. gasseri, L. gastricum, L. salivaris.* | *Martín et al. (2012)* |
| Spanish | CS | 23 | 7, 14, 35 | qPCR, culture | *Staphylococcus epidermidis, S. aureus, Staphylococcus* spp., *Enterococcus faecalis, E. faecium, Streptococcus.* | *Staphylococcus epidermidis, S. aureus, others Staphylococcus, Enterococcus, Streptococcus* | *Jiménez et al. (2008)* |
| Irish | LG | 10 | 7–14 | Illumina/(V3–V4) | *Pseudomonas, Staphylococcus, Streptococcus, Elizabethkingia.* | *Bifidobacterium, Gardnerella* sp., *Bacteroidetes.* | *Murphy et al. (2017)* |
| Swiss | CS | 21 | 3–6, 9-17 | Pyrosequencing | *Pseudomonas, Ralstonia, Streptococcus, Staphylococcus.* | *Bifidobacterium, Bacteroidetes, Parabacteroidetes* | *Jost et al. (2014)* |
| Italian | CS | 8 | 90 | Illumina/Metagenome | *Corynebacterium, Kroppenstedtii, Staphylococcus epidermidis.* | *E. coli, Bifidobacterium, Veillonella, Bacteroides.* | *Asnicar et al. (2017)* |
| Italian | CS | 36 | 2–3 (milk), 20 (stool) | Illumina/(V3–V4) | Streptococcaceae, Paenibacillaceae, Lachnospiraceae, Bifidobacteriaceae. | *Bifidobacterium,* Enterobacteriaceae, Streptococcaceae, Bacteroidaceae. | *Biagi et al. (2017)* |
| African (Kenya, Ethiopia | CS | 377 | 30–90 | Illumina/(V1–V3) | *Corynebacterium, Streptococcus.* | *Veillonella, Lactobacillus.* | *Lackey et al. (2019)* |

**Notes.**

Abbreviations: n, number of samples; TS, type of study; CS, Cross sectional study; LG, longitudinal study; D, days after delivery where samples were taken; Method, method of analysis of 16S rRNA gene.
**Table 3  Reports of function of most abundant taxa found in this study in both, human milk and neonatal stool.**

| Taxa | Immune function | Reference | Colonization and metabolism | Reference |
|---|---|---|---|---|
| | Associated with growth inhibition and spread of *S. aureus*. | *Shu et al. (2013)* | Involved in early gut colonization in breastfeed infants. Best natural producer of propionate and lactate. | *Rocha Martin et al. (2019)* |
| *Propionibacterium* | *Propionibacterium* mitigates intestinal inflammation via Th17 cell regulation and maintenance of T-cells and IL-10 in infants fed with breast milk. | *Colliou et al. (2017)*, *Morrow et al. (2013)* | Prepares the gut environment for late colonizers such as *Faecalibacterium* and *Coprococcus*, which depend on the presence of SCFA. | *Morrow et al. (2013)* |
| | Protective factor against the development of necrotizing enterocolitis in preterm infants. | *Colliou et al. (2017)* | | |
| Sphingomonadaceae | Modulation and maintenance of the immune response. | *D'Auria et al. (2013)* | Colonizer of the breast ductal system and mammary tissue. | *Chan et al. (2016)* |
| | Potent stimulator of NK cells and cytokine release through its glycosphingolipids. | *Long et al. (2007)* | Ability to degrade aromatic hydrocarbons mainly associated with breast cancer. | *Urbaniak et al. (2014)* |
| | Promotes the protection against chemically induced colitis through the development of FOXP3+ T Reg cells in mice. | *Atarashi et al. (2011)* | Associated with carbohydrate metabolism by pentose metabolism. | *Cynkin & Delwiche (1958)* |
| Clostridiaceae | Protection against IgE-mediated disease. | *Kamada et al. (2013)*, *Morrison & Preston, 2016* | Butyrate producer. | *Morrison & Preston (2016)* |
| | Promotes the generation of Th17 cells in mice by stimulating IL-6 and IL-23. | *Atarashi et al. (2011)* | | |
| *Staphylococcus* | Development of the neonatal immune system. | *Lundell et al. (2009)* | First colonizer of the gut tract in the first month by overexpression of adhesion-related genes. | *Martín et al. (2012)* |
| | Super antigen function stimulates the systemic secretion of IgA in neonates, protecting against allergies. | *Martín et al. (2012)* | Ability to degrade high concentration of oligosaccharides in human milk. | *Duncan et al. (2002)*, *Urbaniak et al. (2016)* |
| *Bifidobacterium* | The pili and extracellular polysaccharides promote the modulation of the infant immune system without causing an adverse inflammatory response. Induction of T-reg cells via butyric acid and propionic acid in mouse models and cell lines. | *Turroni et al. (2010)* | Exceptional capacity to participate in the saccharolytic fermentation of carbohydrates, which end −products that positively affect host cells and gut bacterial community. | *Tanaka & Nakayama (2017)* |
| | Decrease the incidence of allergies. | *Bottacini, Van Sinderen & Ventura (2017)* | Early gut colonizer, with high capacity to persist and to colonize. | *Turroni et al. (2010)* |
as *Staphylococcu* s (*Hunt et al., 2012*), while others like *Propionibacterium, Staphylococcus,* and *Streptococcus* use lactose as energy source (*Chassard, de Wouters & Lacroix, 2014*). These bacteria, have also capacity to metabolize lactate to propionic acid, conditioning the gut environment for later colonizers as *Faecalibacterium, Coprococcus,* and *Roseburia,* which rely on the presence of short chain fatty acids (SCFA) for growth (*Duncan et al., 2002*). On the other hand, in our study some of the abundant bacterial taxa found in the human milk such as Rhodobacteraceae, Sphingomonadaceae and Phyllobacteriaceae families, were observed in low abundance in the corresponding stool; suggesting these bacteria are likely not abundant colonizers of the distal colon, at least in our samples. We believe members of these families may have a transient presence in this environment or they may reside in other anatomical parts of the GI tract. We hypothesize that the presence of these taxa is important for immunological stimulation during the early days of colonization.

In Mexico, mothers habitually do not thoroughly cleanse their breasts before breastfeeding. For this reason and because we wanted to know the bacterial composition transferred from mother to neonate during the process of breastfeeding, in our study the mothers did not clean their breast prior to sample collection. There are other published studies on the characterization of "Breastfeeding-associated microbiota" where the human milk was collected without an aseptic cleaning procedure (*Ward et al., 2013*; *Urbaniak et al., 2014*; *Sakwinska et al., 2016*). Based on our results, we think that in addition to intrinsic bacteria from the human milk, skin–associated taxa are an additional source of bacteria in the studied neonates. On the other hand, the microbial source tracker analysis indicated that breastfeeding microbiota was the main source of bacteria in most of the fecal samples of neonates, where only eight samples showed a total predominance of bacteria identified as "unknown source" (Fig. 6A and Fig. S6). The source of these "unknown" bacteria may be the maternal gastrointestinal tract and the intrauterine environment such as placenta and amniotic fluid, which are mainly dominated by members of the orders Clostridiales, Bacteroidales, Lactobacillales, and Enterobacteriales (*Parnell et al., 2017*). Likewise, the delivery mode could also contribute to the abundance of these taxa. The fetal gut is exposed to these bacteria because large quantities of amniotic fluid are swallowed during the last stage of pregnancy (*Neu & Rushing, 2011*); in fact, a recent study showed a high degree of similarity between meconium bacteria and those found in amniotic fluid (*Ardissone et al., 2014*). We can speculate that this microbial profile changes gradually after the birth due to the incorporation of new bacterial members mainly transferred through breastfeeding.

The delivery mode has been one of the most studied perinatal factors due to its potential effect on milk microbiota, as well as on the neonatal gut microbiota composition. Studies conducted in Spanish, Italian (*Khodayar-Pardo et al., 2014*; *Cabrera-Rubio et al., 2016*; *Toscano et al., 2017*) and Chinese (*Li et al., 2017*) populations, have reported significant differences in milk microbial profiles between mothers who delivered vaginally and those who delivered by C-section. Conversely, in a report by Urbaniak and coworkers (2016), no differences were identified in a Canadian population. In our study, we did not find association between human milk microbiota composition and the delivery mode, which can be explained because in our cohort we had only women with non-elective C-sections;

the reported changes in the milk microbial communities appear to be more pronounced in women undergoing elective, than non-elective C-sections (*Cabrera-Rubio et al., 2012*). The same group also reported that milk microbiota of women who gave birth by non-elective C-section was comparable to women who delivered vaginally (*Cabrera-Rubio et al., 2016*). They suggest that physiological (e.g., hormonal) changes produced in the mother during the labor process, may influence the composition of the bacterial community. On the other hand, in our work the delivery mode had an impact on the infant gut microbiota composition. C-section delivery was associated with larger infant gut microbiota diversity and richness. Unlike our findings, most reports have shown lower diversity and abundance of gut microbiota during the first month of life in newborns born by C-section (*Penders et al., 2006*; *Lee et al., 2016*). Regarding *Staphylococcus*, *Propionibacterium*, *Clostridium*, and *Corynebacterium* genera, several authors have found higher abundance in meconium obtained from C-section neonates in accordance with our findings (*Dominguez-Bello et al., 2010*; *Liu et al., 2015*). The abundance of *Bacteroides* and *Bifidobacterium* genera on the other hand, has been reported to be decreased in C-section delivered infants (*Montoya-Williams et al., 2018*). In accordance with this, we observed a tendency of depletion for these genera in the C-section stool samples of this work, but without statistical significance. Our results suggest that the delivery mode markedly modulates the gut microbiota composition in the newborn from the first days of life, being exposed to bacteria from the mother's skin and vaginal canal microbiota, as well as non–maternal sources at the hospital environment.

The metagenomic predictions analyses of our data disclosed higher abundance of bacterial metabolic pathways related to fatty acids metabolism, peroxisomes, and PPAR signaling pathways, in human milk in comparison to neonatal stool samples. Lipids provide the major portion (45%–55%) of the total energy content of human milk, contributing to up to 90% of the energy required by exclusively breastfed infants during the first 6 months of life (*Brenna & Lapillonne, 2009*; *Koletzko et al., 2011*); with the peroxisomes being one of the main specialized cellular organelles where fatty acid metabolism occurs (*Wanders, Waterham & Ferdinandusse, 2016*). Therefore, the high content of lipids present in human milk, could not only depend on the mother's endogenous mammary alveolar fatty acids synthesis, but also it may depend on the contribution of short chain fatty acids as acetate and lactate, produced by some bacteria of the milk microbiota (*Henrick et al., 2018*). It is known that butyrate and propionate participate not only in the regulation of lipid metabolism, but also have a role in the regulation of immune responses and inflammation through activation of the peroxisome proliferator-activated receptors (PPAR) (*Nepelska et al., 2017*; *Hasan, Rahman & Kobori, 2019*).

Neonates showed a high abundance of metabolic pathways in the gut microbiota related to carbohydrate metabolism. Several studies have reported that Bifidobacteria—whose abundance is significantly larger in our neonate samples—can metabolize a wide variety of carbohydrates, such as lactose, the main sugar of human milk (Table 3). In addition, Bifidobacteria can also degrade lactate anaerobically to pyruvate, to generate energy through the pentose phosphate pathway (*Wolfe, 2015*). This metabolic adaptation of Bifidobacteria to sugar–rich environments such as human milk, is due to its 5.5% of genomic sequences associated to the metabolism of carbohydrates (*Milani et al., 2015*). Likewise, in the

neonatal stool, an abundance of genes involved in the vitamin B6 (pyridoxine) metabolism was detected. The production of B-vitamins has been also associated to bacteria such as *Bifidobacterium* and *Acinetobacter* (*Magnúsdóttir et al., 2015*). Beyond its role as a necessary cofactor in the folate cycle, vitamin B6 also plays an important role in amino acid metabolism, the synthesis of neurotransmitters and the hormone melatonin (*Rossi, Amaretti & Raimondi, 2011*). Metabolic pathways involved in bacterial colonization and proliferation, such as bacterial secretion system and bacterial chemotaxis as well as antigens presentation, were also over-represented in neonatal stool. Besides the interaction of bacteria with the host, these functions are essential for the gradual colonization of the immature neonatal gut. Adhesion to the intestinal mucosa is an important feature for bacteria who colonize the gut (*González-Rodríguez et al., 2013*). Our study provides the first results on human milk microbiota obtained under physiological conditions in healthy Mexican mothers, and its association on the early colonization of the neonatal gut, as well as the effect of delivery mode on the human milk and neonatal stool microbiota.

This study has the following limitations with respect to the environmental variables, we could not obtain the maternal information related to dietary habits during pregnancy, which could be relevant to the human milk microbial profiles. Although the core microbiome and microbial source trackers analysis showed evidence about the shared bacterial taxa and the vertical transfer of the microbiota from mother to child, the presence of common taxa does not necessarily validate the vertical transmission, since the species could a have a different origin, or identified taxa in different niches (human milk and neonate gut) not necessarily are the same. Therefore, this result must be confirmed evaluating whether shared bacteria belong to the same strain. Similarly, the origin of these bacteria is equally important, especially to know which strain comes from the mother gut and which come from the breast skin, as well as to determine the magnitude of these contributions in the gut colonization of the newborn. However, we neither collected stool nor breast skin samples from the mother. Additionally, there is a possibility that the specialized kit method used to extract DNA from each type of sample (human milk, stool) affects the microbial composition.

## CONCLUSIONS

Our study provides evidence that the human milk is one of the main sources of bacteria that colonize the neonatal gastrointestinal tract from the first days of life. This gut colonization is characterized by a high dominance of bacterial taxa, mainly by members of the phyla Firmicutes (*Clostridium gasigenes*, *Streptococcus*, *Staphylococcus*, and *Enterococcus*), Proteobacteria (Pseudomonadaceae, Sphingomonadaceae, and Bradyrhizobiaceae), Actinobacteria (*Bifidobacterium*, and *Propionibacterium acnes*) and Bacteroidetes (*Bacteroides*), which may be transferred through lactation, while other taxa in minor proportion such as Clostridiales could come from other sources. Likewise, perinatal factors such as delivery mode suggest an association with the gut microbiota composition in the neonates. Finally, we found a higher abundance of predicted bacterial metabolic pathways associated with lipid metabolism in human milk, while in the neonates

the functional pathways are more associated to carbohydrate metabolism and bacterial colonization. This study contributes to the knowledge on human milk and neonatal stool microbiota in healthy Mexican population and supports the idea of vertical mother-neonate transmission through exclusive breastfeeding.

## ACKNOWLEDGEMENTS

We are grateful to Flor María Galván-Rodríguez, Carolina Miranda-Brito, Francisco Guillermo Borques-Arreortua and Rodrigo García-Gutiérrez for support in the lab work and in sample collection; Alma Lemus-Hernández, and Viridiana Rosas-Ocegueda for administrative assistance; and Tania Smith Marquez for English edition of first draft. The authors are deeply indebted to all the participant mothers and babies of this study.

### Funding

This work was supported by Cinvestav, CONACyT-163235 INFR-2011-01, Fondo SEP-Cinvestav-2018-174, CONACyT Post-Doctoral Fellowship 398875 (IGG), and CONACyT Doctoral Fellowship 777953 (KCC). Ciudad Sen S. de R. L. de C. V. provided funds for the publication fee. The funders had no role in study design, data collection and analysis, decision to publish, or preparation of the manuscript.

### Grant Disclosures

The following grant information was disclosed by the authors:
Cinvestav.
CONACyT-163235 INFR-2011-01.
Fondo SEP-Cinvestav-2018-174.
CONACyT Post-Doctoral Fellowship: 398875 (IGG).
CONACyT Doctoral Fellowship: 777953 (KCC).

### Competing Interests

The authors declare there are no competing interests.

### Author Contributions

- Karina Corona-Cervantes and Igrid García-González conceived and designed the experiments, performed the experiments, analyzed the data, prepared figures and/or tables, authored or reviewed drafts of the paper, and approved the final draft.
- Loan Edel Villalobos-Flores and Fernando Hernández-Quiroz analyzed the data, prepared figures and/or tables, and approved the final draft.
- Alberto Piña-Escobedo performed the experiments, authored or reviewed drafts of the paper, prepare sequencing data files, and approved the final draft.
- Carlos Hoyo-Vadillo conceived and designed the experiments, authored or reviewed drafts of the paper, supervising activities, and approved the final draft.
- Martín Noé Rangel-Calvillo conceived and designed the experiments, authored or reviewed drafts of the paper, and approved the final draft.

- Jaime García-Mena conceived and designed the experiments, authored or reviewed drafts of the paper, and approved the final draft.

## Human Ethics

The following information was supplied relating to ethical approvals (i.e., approving body and any reference numbers):

The protocol was approved by the Ethics Committee of the General Hospital ''Dr. José María Rodríguez'' (Project identification code: 217B560002018006).

## Data Availability

The sequence and corresponding mapping files for all samples used in this study are available in the NCBI BioSample repository: PRJNA548324.

## Supplemental Information

Supplemental information for this article can be found online at http://dx.doi.org/10.7717/peerj.9205#supplemental-information.

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
