# Peer review of "Human milk microbiota associated with early colonization of the neonatal gut in Mexican newborns"

_PeerJ, doi:10.7717/peerj.9205_

## Round 0.1 · original submission · Major Revisions

Dear Dr. Corona-Cervantes and colleagues:

Thanks for submitting your manuscript to PeerJ. I have now received three independent reviews of your work, and as you will see, the reviewers raised some concerns about the research. Despite this, these reviewers are optimistic about your work and the potential impact it will have on research studying human milk microbiota associated with newborns. Thus, I encourage you to revise your manuscript, accordingly, taking into account all of the concerns raised by both reviewers.

While the concerns of the reviewers are relatively minor, this is a major revision to ensure that the original reviewers have a chance to evaluate your responses to their concerns. There are many suggestions, which I am sure will greatly improve your manuscript once addressed.

Importantly, please ensure that an English expert has edited your revised manuscript for content and clarity. Please also ensure that your figures and tables contain all of the information that is necessary to support your findings and observations.

I look forward to seeing your revision, and thanks again for submitting your work to PeerJ.

Good luck with your revision,

-joe

Reviewer 1 ·

Basic reporting

a. There are several instances as described below in the Genereal comments in which the sentence structure is awkward or needs to be edited to be easily read.
b. Intro and background provide sufficient context.

Experimental design

a. There are just a few questions in regards to the methods but with some revision, there is sufficient detail and information.

Validity of the findings

a. The authors have provided a report on a study that examines the milk microbiota and infant fecal microbiota in a self-reported healthy Mexican cohort of mother/infant dyads during the first 6 days postpartum/infant life.
b. The authors provide speculation about the roles milk bacteria play in the early colonization of the infant’s gastrointestinal microbiome.
c. Caution, however, needs to be heeded in not using words that reflect causality and instead describe results as associations/correlations.
d. In the conclusion, I think it would be better to say that they have provided evidence rather than “confirmed” that milk is the main source of bacteria that colonize the infant’s GI tract.

Additional comments

Abstract

Ln 20: Would recommend changing the word “gut” to “gastrointestinal tract” throughout the manuscript.

Line 20-22: Sentence reads awkwardly. Maybe change “Even though” to “However”?

Ln 25: In abstract, don’t need the use of “The” in the “The human milk…”. Actually, this is common throughout manuscript and there are several instances where this needs to be edited.

Line 27: “dominated the human milk bacterial community.” Did these bacteria actually “dominate” the community or were they more “prevalent”? or both?

Ln 29: Would recommend changing the word “breast” in “breast milk…” to “human” throughout manuscript.

Ln 29: How was neonatal gut microbiota associated with delivery mode? With microbial diversity?

Ln 44: please check to see whether ‘in utero’ should be italicized.

Ln 52-58. This is a very long sentence. To help with the readability of this sentence, please consider creating two sentences, perhaps by separating in line 53.

Line 63: change “to” to “with”

Line 72: “microbiota” is a noun; perhaps better to use “microbial”

Line 77-80: There have also been studies in the USA by this group (Hunt et al. 2011, Williams et al. 2017 and 2019, Lackey et al. 2019)


Methods

Line 109: How long after the infant was fed? Immediately following?
Line 112: How were feces sampled? Using a sterile scoop, spatula, or swab of some sort? How soon after infant defecated were samples collected?
Line 118 and 120: What temperature was milk centrifuged at?
Line 124: “QIAmp” should be “QIAamp”.

Line 126: Negative controls were used during the DNA extraction but were they also sequenced?

Line 136-138: Should be two sentences or separated by semicolon before “nuclease free sterile water was used.”

Line 140: How were PCR amplicons quantitated? Or were equal volumes pooled?

Line 165: Might be better to describe Chao 1 as a richness estimator


Results

Line 220: For neonate description, would be helpful to separate out female and male by delivery mode.

Line 228-230: When I look at Figure S1, I see a continued upward trend as the read depth goes out to 20,000 reads. Perhaps I am misinterpreting what the authors are meaning when they say 96.7% of total samples did not show new Observed_otus?

Figure 1:
- “Human” and “Neonatal” should start with lowercase letters

Line 234: “being this the only phylum with statistically significant difference” seems awkward in the sentence.

Figure 3:
- Next to last sentence, “H” needs to be capitalized in “human”, plus missing a space to separate sentences.

Line 275: “Only the Shannon and Simpson indexes showed higher…” This is confusing since the sentence before says that richness was higher in the milk microbial community. But then the next sentence says with BH correction, there were no significant differences.

Table S5: Something doesn’t seem right in this table. Both observed and Chao 1 richness estimators are around 60-75 but in looking at the rarefaction curves in Figure S1, the average seems like it would be greater than that. Also, in looking at 3A, the median or mean line is much higher.
Additionally, I would also like to see the SD or SEM for the values in Table S5.
There is also something odd about the high values for Shannon and Simpson. They do not match the boxplots in 3A as well.
I am guessing that the alpha diversity indices were calculated at the otu level, but even as such there is discrepancy between Figure 3A and the table.

Additionally, were the sample read counts rarefied or normalized somehow prior to calculation of alpha diversity? Read depth seemed to be quite varied across samples and read depth can have a significant impact of diversity indices.
If read counts were not rarefied, did the authors check to see if read depth contributed to the overall community structure and result in biases in diversity analysis? Were there differences in read count by sample type or delivery mode?

Line 279: There does appear to be grouping but there is not a “clear separation. It would also be helpful to the reader if the colors were consistent between Fig3A and 3B for human milk and stool groups.

Figure 3B – Is this a different ordination plot than Figure S2? The PC axes have different percentages. And if different, what is Figure S2 displaying? How does this provide “more precise information”?

Line 280: Isn’t Adonis a PERMANOVA? Did the authors mean anosim instead of Adonis test for similarity?

Line 286: Delivery mode is associated with bacterial richness and diversity. The word “influenced” suggests causality and thus, should not be used.

Line 288: Once again, there appears to something off about the values in Table S6. Why are these alpha diversity values so different from Table S5 which is also different from Figure 3A? They also seem quite different from Figure 4.

What is the difference between S3A and S3B? I did not find the legend for the Supplemental figures.

Line 300-302: Although evaluating relationships with maternal age is interesting, what about time postpartum? Human milk composition undergoes a lot of compositional change over the first week of lactation. This would be of great interest to evaluate as well.

In addition to maternal age, what about testing for relationships with parity? And this information, i.e. description of parity for the population, should also be included in Table 1.

Line 315: Table S7, it might just be alterations in the table due to a different computer, but the columns should be widened so the text in a word doesn’t wrap to the next line.
Also, the row of Bacteroidetes has an extra column between “Genus/species” and “Human milk”

I also don’t understand what the OTUs count represents in Table S7. Are these the number of reads attributed to each genus or species? Is it the total read count across all samples?
What are these numbers?

Line 316: I think this sentence is missing a “that” or “which” before the “originated from the human milk”.

Line 318 (Figure S6): I wonder what associations are present between mode of delivery and amount of contribution from milk microbiota. And is the day postpartum associated with increasing or decreasing contributions of milk bacteria to the infant microbial community? Did the authors examine these factors for association with amount of milk bacteria that contributes to infant fecal microbial community?

Line 319: “came from other unknown source no characterized in this work” does not make sense the way it is worded; needs to be edited.

Line 320-323: Not clear what this sentence means. Is it saying that the bacteria in the fecal samples that were identified as coming from an unknown source and not human milk were mainly taxonomically identified as being from the Clostridiales order? Maybe what is missing is “The taxonomic analysis of the bacteria in the fecal samples…”

Figure 6B: Bar label needs to be corrected to “Unknown Source Group” by adding an ‘n’ at end of “Unknow”.

Line 325: “Predicted” should not be capitalized.

Table S8: Should the differences between means in the table correspond to the right side of Figure 7? The values don’t seem to match up. In addition, Table S8 discusses q-values but there is not a column for the q-value.
Under ‘Corrected p-value’ column the “-3” is superscripted on all except one value. Please make consistent.
Spacing in “Antigen processing and presentation” is different in this cell as compared to others.


Discussion

Line 343 (Table 2): Under the ‘Method’ column, to be consistent, shouldn’t the Asnicar reference be “Illumina/Metagenome”

Line 347: It is unclear in Hunt et al. 2011 which, if any, milk samples were collected during the first 30 days postpartum. According to Williams et al. J Human Lact 2017, which utilized the same milk samples, the average time postpartum was around 5 mo.

Line 359: “Chan el al., 2016” needs to be edited to be ‘Chan et al., 2016’

Line 368-371: This sentence doesn’t quite read correctly, i.e. the section on “core microbiota” added of a pan-microbiota”

Line 389: Although the references appear to be in chronological order, it might be more helpful to the reader to have the references in the order of the populations or conversely, put the populations in order of the references.

Line 406: Which bacteria are being referred to by “its” in “its ability to use” HMO? I don’t think the bacteria mentioned above utilize HMO. I think the Hunt et al. 2012 article describes Staphylococcus increasing in concentration but not actually utilizing the HMO. The Yu paper describes Bifidobacteria utilizing HMO. I also don’t know that the Rocha Martin manuscript provides evidence of use of fatty acids as energy sources. I didn’t do a thorough read but at first glance this reference doesn’t seem to describe this.

Line 414-417: What about the possibility that these bacteria are in other parts of the GI tract?

Line 419-420: “We try” seems awkward; maybe “We wanted to replicate” would read better?
This whole sentence needs to be reworded as the sentence construction is awkward.

Line 425-429: In this study, both bacteria from skin and bacteria from within the mammary gland would be present in the milk sample. Thus, bacteria in skin would be included in the milk microbiota observed in source tracker.

Line 448-450: This sentence reads awkwardly.

Line 460: “et la.” should be “et al.”

Line 464: Did all of the vaginal births occur at home? If not and some of the vaginal births occurred in the hospital as well, delivery mode and site of birth are potentially two different factors.

Line 471: maybe change “being the peroxisomes…” to “with the peroxisomes being one of the main …”

Line 473: change “mother” to “mother’s”

Line 482: significantly increased from what? Or compared to what? Milk bacterial communities?

Line 499, 524: Once again, would caution against using the word “influence(s)” and instead use words that reflect that these are associations and/or correlations.


References

There are several citations in which the journal titles need to be italicized. Additionally, the sentence case format is not consistent across references. And there are some references that have capitalization of some words in the middle of the sentences.
Here are just a few examples:
Line 798: odd capitalization of “Are” in the middle of the title.
Line 848: “Normal Pregnancy”

Line 805, 892, 924: need line space between references.

·

Basic reporting

The manuscript submitted by Corona-Cervantes et al. describes the structure and, to a limited extent, predicted genetic capacity of bacterial communities detected in maternal milk samples and infant faecal samples from a Mexican cohort.
The language used is of a good standard and is clear and unambiguous. I would guess that English is not the main authors first language and there are minor points throughout the manuscript that might have been worded slightly differently by a native English speaker, but these differences do not impact on the legibility of the manuscript (for example, lines 97/98: 'Apgar score greater than 7 at minute 5 after born.' could be worded as '...at 5 minutes after birth.' Also line 462: 'markedly modules' could be 'markedly modulates'.)
The structure of the manuscript is good. Figures and tables are appropriate and well presented (having said that, Tables 2 and 3 feel like they belong in a review rather than a research paper, but they do work well in this study). The study includes appropriate literature and background/context. The results and hypothesis align and raw data is appropriately archived (and available).

Note: Fig 3 legend. Penultimate sentence. Insert a space after the full stop and add a capital h for ‘Human…’

Line 408: There appears to be a 'stray' reference [48].

Line 460: Dominguez-Bello et al (not et la).

Experimental design

The research question is defined and given the possibility of different microbial communities within different ethnic/cultural groups (with unique dietary and behavioural traits) it is relevant to carry out studies that focus on previously under or not-reported cohorts.
The methods used are appropriate for the study and appear to be technically well done. Method details are suitable.

Further comments below:

The materials/methods noted that Blank extraction controls were carried out - were they negative? If not, how was environmental contamination controlled for?

It appears that two DNA extraction methods were utilised for milk and faecal samples. Neither appear to include physical disruption (some kind of physical disruption is often recommended, particularly for bacteria such as bifidobacteria. For example see: Walker AW, Martin JC, Scott P, Parkhill J, Flint HJ, Scott KP. Microbiome. 2015 Jun 22;3:26. doi: 10.1186/s40168-015-0087-4. eCollection 2015.
16S rRNA gene-based profiling of the human infant gut microbiota is strongly influenced by sample processing and PCR primer choice.

Could any of the differences, particularly diversity, between the milk and faecal microbiomes be due to the different methods used to extract genomic DNA?

Was there a rationale behind the use of qiime v1.9 rather than the updated qiime2? The updated package includes sequence error correction algorithms (such as dada2, which incorporates a ‘pyro’ option for analysing PGM data). Error correction can reduce the 'tail' of 'rare species' and thus have an impact on diversity measurements. However, as long as all samples are analysed using the same methods comparisons should still be valid (as is the case for this study).

Fig 3: Does significance in the two alpha diversity metrics that include components of richness and evenness suggest that much of the difference in diversity between the milk and faecal samples is due to higher evenness in the milk samples? This could be tested by running Pielou’s evenness test in qiime.

Is there a rationale behind the use of different ‘types’ of figure when describing alpha diversity between milk/faeces groups and vaginal/c-section groups? Also, why include Simpson and Shannon indices only for the milk/faeces comparison?

Fig 5. What does the tree/cladogram next to the heat map represent? Is the clustering based on average relative abundance?

Obviously it is extremely difficult to design and carry out microbiota-based experiments with high power to detect differences between groups (given inter-individual variation). But, were any power calculations carried out prior to commencing the study?

Why use syber green mixes for amplification prior to sequence library preparation? Might it have been better to use a high fidelity taq?

I was not aware that FastQC could be used to trim reads to a set length (I've always used this program to carry out quality control analyses). The Babraham Institute (who created FastQC) also supply 'Trim Galore' for sequence trimming.

Validity of the findings

The discussion notes low relative abundance of Bifidobacteria and Lactobacillus in comparison with other studies. Could this be an impact of the DNA extraction methods chosen for the current study? Indeed a lack of standardised methods makes comparisons between studies quite difficult at the best of times.

The study conclusions are appropriate and well supported. There is some speculation, and this is identified.

Overall the study adds to the knowledge of milk and infant faecal microbiota composition (particularly for a previously unreported study group) and provides valuable information relating to the colonisation of the human gut.

Reviewer 3 ·

Basic reporting

1) In general, the first half of the paper does not focus on the main Aim (sharing of milk/gut taxa). I recommend focusing on your main Aim and most novel findings, rather than descriptive results and associations that are already well known and not your main purpose.

2) Introduction: This was hard to follow at times. The first paragraph does not effectively present the current theories for the origins of milk bacteria. While the main theories are both mentioned (entero-mammary pathway and exogeneous origins) they are presented separately in a way that makes the paragraph contradict itself. Also, the ‘prebiotic’ function of milk (via oligosaccharides) is not mentioned. Midway through the second paragraph, the origins of milk bacteria are discussed again – this was already discussed in the previous paragraph and should be moved or omitted. In the final paragraph, the statement of the study Aim does not mention sharing of milk/gut bacteria. Isn’t this the primary Aim?

3) In general, the discussion is quite long and could be condensed substantially. Also, please revise the opening paragraph of the discussion to clearly state your main findings.

Experimental design

4) Authors do not address the well-known contamination problem associated with low biomass samples. They mention that negative controls have been added during sequencing, however there is no mention on whether any kit contaminants were identified in the negative controls. The authors should reanalyse their data and discard any OTUs identified in the negative control and in the samples or use a different accepted method for decontamination using their negative controls, such as the use of the decontam package in R.

5) Other methodological issues:

a) Line99 Do mothers delivering by c-section receive antibiotics in Mexico? (Antibiotic use is mentioned as an exclusion criterion – please clarify.)
b) Line115 Were samples really processed for DNA extraction within 24 hours of collection? Normally samples are processed in batches (or ideally, all at once) to reduce variability. Please clarify and explain.
c) The authors have shown a detailed description of laboratory methods, however an explanation of what “equal amounts” of each barcoded amplicon means [in Ln 120] is needed. Ideally, this should be an equal mass rather than volume of each sample/amplicon to avoid biases in sequencing (Particularly since stool and milk usually have very different DNA concentrations).
d) Authors have used an older version of QIIME that has been depreciated as of January 1, 2018. I would recommend reanalysing with the most recent version of QIIME2.
e) Normalization methods are not described. The sequences generated belong to different sample types with very different microbial biomass. This needs to be taken into account when pre-processing the sequences, calculating relative abundances and correlation analyses.
f) Authors mentioned FastQC was used to trim the sequences of low quality. As far as I am aware, fastQC is a QC tool that just estimates the quality of sequences and does not have any trimming parameters. Could the authors please confirm what tool was used for trimming of the sequences?
g) Ln 230: It is unclear from the authors description whether rarefaction was performed (and to what depth) or if sequencing depth was accounted for using a different method. Accounting for this is particularly important with variable sequencing depths, and the rarefaction curves in Fig S1 indicate to me that samples were variable in sequencing depth. Also, for a number of samples, the rarefaction curves do not reach a plateau which does not support the authors’ statement that “The rarefaction curves indicate that depth of sequencing coverage was acceptable” (Line228)
h) Ln 199: The authors mention both the use of PERMANOVA and Adonis, but both are the same statistical analysis (permutational multivariate ANOVA). Either PERMANOVA was performed using the Adonis function in R or using some other program. Since they are the same analysis, ideally only one should be reported. [Adonis is the name of the function for this analysis used in the vegan package in R, while PERMANOVA is the general acronym for the analysis (Anderson, M.J., Wiley StatsRef: Statistics Reference Online, 2014, pp. 1-15)]
i) Ln 168: Euclidean distance is separate from UniFrac distance (you cannot do a UniFrac analysis with Euclidean distance) – Recommend clarifying which was used for beta-diversity.

Validity of the findings

6) The conclusion is too strong. This study is inferring the source of bacteria bioinformatically using 16S (not strain-level) data, without analysing any additional/alternative sources. It is important to acknowledge that this study cannot prove or quantify the ‘vertical transfer’ of bacteria from breast milk to the infant gut, and that the co-occurrence of similar bacteria could also simply reflect colonization of mother and infant by the same bacteria from their shared environment.

7) Using the same method (SourceTracker), Pannaraj et al. estimated that 28% of infant gut bacteria originated from breast milk. The current study estimates 68%. This is a very large difference. Please discuss.

8) It is very well known that method of delivery influences the infant gut microbiome. One of the strongest and most consistently identified differences is the depletion of Bacteroides taxa in CS-born infants. It is strange that this was not found in the current study. Please confirm and comment on this.

9) Figure 3b consists of 3 PCoA plots, where the authors explain a clear difference in the 2nd plot of 3b. Do the other 2 PCoA plots explain any other variance between the samples? the authors do not explain the other PCoA plots.

10) The results paragraph starting at L269 and related TableS5: only p-values are reported; this is not very meaningful. Please report effect sizes and include (especially in the table!) a measure of variation.

Additional comments

Corona-Cervantes and colleagues used 16S rRNA sequencing of the V3 region to profile the microbiota of human milk and neonatal stool from 67 Mexican mother-infant dyads. Similar larger studies have been performed, albeit in different settings. The authors claim that their results indicate 68% of bacterial taxa from the infant gut originate from breast milk and interpret this as evidence of vertical transfer, stating that their study “confirms that breast milk is the main source of bacteria that colonize the neonatal gastrointestinal tract from the first days of life.” I think this statement is probably too strong. Overall, the manuscript needs editing for language and flow as it is overly long, and hard to follow in places. I have quite a few questions and concerns about the methodology and presentation of results.

MAJOR concerns are listed in the sections above. Other MINOR concerns are:

11) The term “increase” is misused multiple times when comparing milk to gut data. “Increase” implies a change over time. You are comparing different sample types, so should refer to “higher” or “lower” levels.

12) Line252 I don’t understand this statement; please explain.

13) Line266 You mention “three-fold” but earlier mention that a LDA threshold of 3.5 was used. Please clarify.

14) Fig S2 – please add effect sizes and p-values on the figure.

15) FigS5 – bar graph is not an effective visualization of this data.

16) Fig S6 – large number of pie graphs is not an effective visualization of this data.

17) Do you think that primer differences (you analysed V3, several of the cited studies used V4) might contribute to the different composition you observed?

18) Line389 – please carefully check the cited references here. At least one does not include breast milk analysis.

19) Line 473: do the authors believe that bacteria are metabolically active in the milk (which is continuously produced and ejected frequently), to the extent of influencing short chain fatty acid content?

---

## Round 0.2 · Minor Revisions

Dear Dr. Corona-Cervantes and colleagues:

Thanks for resubmitting your manuscript to PeerJ. I have now received two independent reviews of your work (from the original reviewers), and as you will see, both are still favorable and agree that the resubmission is much improved. Well done! However, one reviewer raised a few more minor concerns about the research, and areas where the manuscript can still be improved.

I agree with the concerns of the reviewer, and thus feel that their criticisms should be adequately addressed before moving forward.

Therefore, I am recommending that you revise your manuscript, accordingly, taking into account all of the issues raised by the reviewer. I do believe that your manuscript will be ready for publication once these issues are addressed.

Good luck with your revision,

-joe

Reviewer 1 ·

Basic reporting

There are still several sentences that would be helpful to restructure or reword for ease of reading.

Experimental design

No comment

Validity of the findings

No comment

Additional comments

I thank the authors for revising the manuscript and providing thorough responses to my questions.

I just have a few additional comments.

Line 126: I thank the authors for providing information to me about the negatives. I think other readers will want to know this information too. Please add in the methods or in the results that since DNA was neither detected by 260 nm absorbance nor a 281 amplicon observed, the negatives were not sequenced.

Line 168: Were samples that had less than 10,000 sequences removed?

Line 320: Not sure having the absolute counts in Table S7 is very helpful; couldn’t these numbers be biased due to high variation in read counts per sample?

Line 414: I do not think the Hunt paper describes Staphylococcus “using” HMO; only that growth increased in the presence of HMO.

There are several instances that the grammar and sentence structure is awkward. I have included a few examples.
Line 354: seems to be missing a few words, e.g. “postpartum in a Canadian population… postpartum in an Irish population”

Line 428-431: This sentence is really long and doesn’t read quite right. Perhaps it would help to split it into two sentences, e.g. “In Mexico, mothers habitually do not thoroughly cleanse their breasts before breastfeeding. For this reason and because we wanted to know the bacterial composition transferred from mother to neonate during the process of breastfeeding, in our study the mothers did not clean their breast prior to sample collection.”

Line 454: “in a report made in Canadian women…” Awkward sentence structure, as the report was not made in Canadian women. “Conversely, in a report by Urbaniak and coworkers (2016), no differences were identified in a Canadian population.”

Line 459: Maybe reword or remove comma to read “The same group also reported…”

Line 461: Start a new sentence at “They suggest that physiological …”

Line 580: extra “B” in title.

·

Basic reporting

The authors can be applauded for answering all the questions of the reviewers and making the necessary changes

Experimental design

The authors have clarified many aspects of the experimental design.

Validity of the findings

The authors have answers all the questions from the reviewers and clarified the findings with more analyses.

Additional comments

no comments

---

## Round 0.3 · accepted · Accept

Dear Dr. Corona-Cervantes and colleagues:

Thanks for once again resubmitting your manuscript to PeerJ. I now believe that your manuscript is suitable for publication. Congratulations! I look forward to seeing this work in print, and I anticipate it being an important resource for groups studying human milk microbiota associated with newborns. Thanks again for choosing PeerJ to publish such important work.

Best,

-joe